# Amazon Wildfires and Respiratory Health: Impacts during the Forest Fire Season from 2009 to 2019

**DOI:** 10.3390/ijerph21060675

**Published:** 2024-05-24

**Authors:** Maura R. Ribeiro, Marcos V. M. Lima, Roberto C. Ilacqua, Eriane J. L. Savoia, Rogerio Alvarenga, Amy Y. Vittor, Rodrigo D. Raimundo, Gabriel Z. Laporta

**Affiliations:** 1Graduate Program in Health Sciences, FMABC Medical School University Center, Santo André 09060-870, SP, Brazil; maura.ribeiro@riobranco.ac.gov.br (M.R.R.); roberto.ilacqua@fmabc.br (R.C.I.); rodrigo.raimundo@fmabc.br (R.D.R.); 2Laboratory of Study Design and Scientific Writing, FMABC Medical School University Center, Santo André 09060-870, SP, Brazil; 3Health Surveillance Department, Acre State Secretary of Health, Rio Branco 69914-220, AC, Brazil; marcos.malveira@ac.gov.br; 4Environmental Health Department, FMABC Medical School University Center, Santo André 09060-870, SP, Brazil; eriane.savoia@fmabc.br (E.J.L.S.); rogerio.alvarenga@fmabc.br (R.A.); 5Department of Medicine, Emerging Pathogens Institute, University of Florida, Gainesville, FL 32611, USA; amy.vittor@medicine.ufl.edu

**Keywords:** air pollution, climate change, epidemiological study, patient admission, respiratory diseases, wildfires

## Abstract

The Brazilian Amazon, a vital tropical region, faces escalating threats from human activities, agriculture, and climate change. This study aims to assess the relationship between forest fire occurrences, meteorological factors, and hospitalizations due to respiratory diseases in the Legal Amazon region from 2009 to 2019. Employing simultaneous equation models with official data, we examined the association between deforestation-induced fires and respiratory health issues. Over the studied period, the Legal Amazon region recorded a staggering 1,438,322 wildfires, with 1,218,606 (85%) occurring during August–December, known as the forest fire season. During the forest fire season, a substantial portion (566,707) of the total 1,532,228 hospital admissions for respiratory diseases were recorded in individuals aged 0–14 years and 60 years and above. A model consisting of two sets of simultaneous equations was constructed. This model illustrates the seasonal fluctuations in meteorological conditions driving human activities associated with increased forest fires. It also represents how air quality variations impact the occurrence of respiratory diseases during forest fires. This modeling approach unveiled that drier conditions, elevated temperatures, and reduced precipitation exacerbate fire incidents, impacting hospital admissions for respiratory diseases at a rate as high as 22 hospital admissions per 1000 forest fire events during the forest fire season in the Legal Amazon, 2009–2019. This research highlights the urgent need for environmental and health policies to mitigate the effects of Amazon rainforest wildfires, stressing the interplay of deforestation, climate change, and human-induced fires on respiratory health.

## 1. Introduction

The Brazilian Amazon is one of the most crucial tropical regions globally and has been facing substantial pressure from human occupation, agricultural expansion, and climate change [1,2,3]. Among the primary activities directly contributing to deforestation in the Amazon are logging and livestock farming with the conversion of forests into pastures for cattle breeding [4,5]. The latter is usually achieved by forest burning through the rudimentary slash-and-burn system [4,5]. Wildfires in the Amazon have always existed, as they are widely used practices in agricultural activities by traditional farmers who lack technologically adapted alternatives [5]. As a result, the dynamics of the forest fires implemented in the 20th century have endured in the Amazon region until today [5].

Forest fires or wildfires in the Amazon are primarily triggered by deforestation fire, which involves clear-cutting the forest, letting it dry, and subsequently burning the trees to prepare the soil for agriculture and livestock [6,7,8]. The effects of deforestation and wildfires in the biome are not only local, but also extend to the regional and global scales with environmental, political, and socioeconomic consequences [6,7]. Prior research has indicated that deforestation, precipitation, and temperature explained roughly 80% of the variability in forest fire seasons, highlighting a positive association between fire count and deforestation [6,7]. The escalation of deforestation since 2012 resulted in a 39% increase in forest fires in 2019, leading to an estimated 3400 additional deaths due to heightened exposure to particulate air pollution [6,7]. Land use and changes over time contribute to global climate changes in various ways, as forests act as natural carbon storage areas, absorbing and storing carbon over time [8,9]. The release of large amounts of carbon dioxide compounds into the atmosphere can cause significant modifications to the Amazon region’s climate, including rising temperatures, extreme weather events, and risks to human health [1,3,10,11,12].

Brazil contributes significantly to the dispersion of global atmospheric pollutants, primarily stemming from deforestation and wildfires [10,13]. Fine particulate matter, such as particles with a diameter of 2.5 μm or smaller (PM_2.5_), is a major component of wildfire smoke [9,14,15]. These tiny particles can penetrate deep into the lungs and pose respiratory health risks [11]. The respiratory tract is the primary organ affected by these atmospheric pollutants, which result in acute respiratory effects associated with recent exposure or chronic effects throughout an individual’s life cycle [11]. The presence of airborne substances harmful to humans represents a significant environmental problem, affecting people in all socioeconomic strata [14,16,17,18]. It both increases the burden of respiratory diseases and the demand for healthcare services, thereby decreasing quality of life and life expectancy [19].

The Federal Constitution of Brazil of 1988 mandated the state to guarantee the right to health for all citizens, and this right extends beyond access to medical care, as environmental conditions are also fundamental determinants of health [20]. Access to clean air, for example, is necessary for the enjoyment of the right to health and the right to life [20,21]. Epidemiological studies have assessed and demonstrated the effects of air pollution on human health as potential factors in the occurrence of respiratory diseases, as well as an increase in the number of hospitalizations [22,23,24,25,26]. Herein, the research problem is that greater numbers of wildfires will contribute to an increase in hospitalizations due to respiratory diseases in children, young adolescents, and older people, because these are the most vulnerable age groups to atmospheric pollution.

The intersection of deforestation, climate change, and human-induced fires has increasingly alarming effects on health [27,28]. This is particularly true and critical to the Amazon region today, which also poses a serious threat to health on a broader scale [16]. This study aimed at to evaluate the association between wildfire incidents, meteorological variables (temperature, precipitation, and relative humidity), and hospital admissions for respiratory diseases in the age groups of 0 to 14 years and 60 years or older in the Legal Amazon states, during the period from 2009 to 2019. The significance of the present study is in its contribution to a better understanding of the effects of Amazon rainforest wildfires on the respiratory health of the local population. This knowledge could have an impact by assisting in the development of environmental and public health policies, implementing intervention measures, and planning programs to control the respiratory diseases associated with environmental pollution.

## 2. Materials and Methods

### 2.1. Study Area and Rationale

The Legal Amazon was established by Law 1806 on 6 January 1953 for its development and control, with subsequent changes in its delimitation through Law 5173 on 27 October 1966 and Complementary Law 124 on 3 January 2007 [29].

All nine states of the Legal Amazon were included in this study, namely: Acre (AC), Amapá (AP), Amazonas (AM), Pará (PA), Rondônia (RO), Roraima (RR), Mato Grosso (MT), Tocantins (TO), and Maranhão (MA) [30]. Despite only 80% of Maranhão being within the official limits of the Legal Amazon, we considered the total state boundaries here to facilitate data extraction and analysis [30].

These nine states correspond to a territorial extension of 5 million km^2^ or 60% of the Brazilian territory. Amazonas has the largest territorial extension, totaling 1.5 million km^2^, and in second place is Pará with 1.2 million km^2^ [30]. The other states have the following territorial areas: Acre (164 thousand km^2^), Amapá (142 thousand km^2^), Maranhão (329 thousand km^2^), Mato Grosso (903 thousand km^2^), Rondônia (237 thousand km^2^), Roraima (223 thousand km^2^), and Tocantins (277 thousand km^2^) [30].

The land use and land cover in each of these Amazonian states have exhibited distinct patterns of change over the years, notably attributed to the transformation of forested areas into pasture within the deforestation frontier macrocosm belt (Figure 1A) [31].

The quantity of forest fire occurrences per municipality, observed between 2009 and 2019, is greater in states where the cumulative deforestation in 2019 reflects a higher conversion of forests into pasture lands (Figure 1B,C). The linear trend of this correlation is as strong as 93% (Figure 1C). At the microscale, we observe that deforestation fire and cattle ranching are intrinsically related to each other. More specifically, pasture lands result from deforestation fires (Figure 1D).

### 2.2. Study Design and Variables

This is an observational, ecological study in epidemiology that employed the data analysis and time series modeling of publicly official data [30,32,33,34] from the nine states comprising the Legal Amazon, spanning from 1 January 2009 to 31 December 2019.

For the data analysis and modeling, we delineated a distinct season of the year: the extended Amazonian summer period [7]. This timeframe aligns with the region’s dry season, characterized by an anticipated low rainfall and elevated temperatures, extending from August to December [35]. This transition period significantly influences the monthly distribution pattern of wildfires and is hereby designated as the forest fire season. The monthly number of cases of hospital admissions (hospitalizations) related to any respiratory diseases, in men or women in the age groups of 0 to 14 years and 60 years or older, per state, August–December 2009–2019, was the response variable in the data analysis and modeling [34]. The monthly number of forest fire incidents, per state, August–December 2009–2019, was the explanatory variable in the data analysis and modeling [32]. The monthly averages of the hourly mean temperature, precipitation, and relative humidity, per state, August–December 2009–2019, were the climate variables in the data analysis and modeling [33].

### 2.3. Sources of Data and Measurement

#### 2.3.1. Hospital Admissions

Hospital morbidity data (hospital admissions) were extracted from the database of the Department of Informatics of the Unified Health System (DATASUS) [34]. The DATASUS system aggregates data from a broad spectrum of healthcare institutions, including both public and private hospitals, ensuring comprehensive coverage of respiratory-related hospital admissions. The system captures essential information such as patient demographics, admission dates, diagnoses, and medical procedures. The dataset encompasses a diverse range of disorders such as asthma, chronic obstructive pulmonary disease, pneumonia, and other respiratory illnesses. With representation across various regions and states within Brazil, the dataset provides a geographically inclusive perspective on respiratory health trends. It is important to note that these data may be influenced by healthcare-seeking behavior, as individuals with varying severities of respiratory conditions may opt for different approaches in seeking medical attention.

The data extraction procedure was carried out by applying specific filters within the hospital morbidity section of the Hospitalization System:General by place of residence—from 2008 onwards;Geographic scope (states in the Legal Amazon);Row (year/month processing);Column (age groups <1 year, 1–14 years, and 60+ years; gender male and female);Content (hospitalizations);Available selections in the International Classification of Diseases-10-chapter X diseases of the respiratory system (J00–J99), listed as follows: Acute bronchitis and acute bronchiolitis;Acute laryngitis and tracheitis;Acute pharyngitis and acute tonsillitis;Asthma;Bronchiectasis;Chronic bronchitis, emphysema, and other chronic obstructive pulmonary diseases;Chronic diseases of tonsils and adenoids;Chronic sinusitis;Influenza;Pneumoconiosis or Pneumonia;Other diseases of the respiratory system.

All these respiratory diseases were chosen and consolidated for the estimation of the response variable of case numbers of hospital admissions or hospitalizations in the data analysis and modeling, per state, August–December 2009–2019.

#### 2.3.2. Forest Fire

The number of wildfire incidents was obtained from the Monitoring Portal for Burnings and Forest Fires of the National Institute for Space Research [32]. Wildfire incidents were identified through remote sensing using the Moderate Resolution Imaging Spectroradiometer (MODIS) sensor on the AQUA_M-T reference satellite [32]. Forest fire incidents were detected through optical telescopes within the infrared energy range (from 0.75 to 1000 µm), with a nominal pixel area of 1 km × 1 km [32].

Data on wildfire incidents were extracted from the reference satellite (AQUA_M-T, MODIS sensor), with a pass in the early afternoon [32]. This satellite was selected for its more regular and stable orbit, contributing to the identification of a representative fraction of the actual number of wildfire incidents for the data analysis. Moreover, as it employs the same detection method and produces images at similar times consistently over the years, we believed it would furnish invaluable data for analyzing the spatial and temporal trends of incidents within the same regions or across regions during periods of interest.

We chose all monthly forest fire events detected by this reference satellite in the Legal Amazon states, covering the period from January 2009 to December 2019. The extracted data were employed to map the number of fire incidents per municipality and state in the Legal Amazon, as depicted in Figure 1B. Additionally, these data were utilized for subsequent data analysis and modeling, per state, August–December 2009–2019.

#### 2.3.3. Climate Factors

Precipitation (mm), temperature (°C), and relative humidity (%) were the meteorological variables extracted from the automatic weather stations belonging to the network of meteorological stations of the National Institute of Meteorology [33].

We selected a total of 141 automatic meteorological stations that are scattered across the nine states comprising the Legal Amazon as follows: Acre (7 stations), Amazonas (19 stations), Amapá (4 stations), Maranhão (17 stations), Mato Grosso (38 stations), Pará (32 stations), Rondônia (4 stations), Roraima (1 station), and Tocantins (20 stations). These automatic stations gather temperature, humidity, and precipitation data on a minute-by-minute basis, providing hourly averages that are publicly accessible on the official website.

We computed the monthly hourly averages of temperature, relative humidity, and precipitation from January 2009 to December 2019 for each state in the Legal Amazon. In states with two or more stations, the monthly hourly average for each parameter was determined by averaging these values across the available stations. In the case of Roraima, the monthly hourly average was calculated using the single available station.

As these meteorological variables exhibited intercorrelations, we utilized principal component analysis to generate two orthogonal axes capable of representing the monthly variations in the meteorological conditions for each state, August–December 2009–2019, for the data analysis and modeling.

### 2.4. Data Analysis and Modeling

In the preliminary data analysis, we examined the linear relationship by conducting Pearson’s correlation test. This allowed us to evaluate the connection between the rate of hospitalizations due to respiratory diseases within the population groups per 10,000 individuals in the overall state population and the total count of forest fire occurrences per state area, August–December 2009–2019. The population and state territory data are accessible through publicly available databases [30]. This analysis posited a connection, considering the reciprocal nature of association between deforestation fires and the incidence of hospital admissions for respiratory diseases. Thus, we proceeded with a more comprehensive approach, incorporating simultaneous equation models that accounted for the meteorological conditions, wildfire incidents, and hospitalizations.

Our model had two sets of simultaneous equations [36]:(1)$$I_{t}=\beta_{1}+\beta_{2}\times {PCA1}_{t}+\beta_{3}\times {PCA2}_{t}+AR(\rho )$$
(2)$$K_{t}=\alpha_{1}+\alpha_{2}\times I_{t}+AR(\rho )$$

Initially, Equation (1) depicts the relationship between the monthly variations in meteorological conditions (precipitation, temperature, and relative humidity) represented by the two axes of the principal component analysis (PCA1 and PCA2) and their impact on forest fires (*I*) in the Amazon, August–December 2009–2019. Following that, Equation (2) investigates the association between forest fires (*I*) and respiratory diseases (*K*). Temporal autocorrelation errors of the second and third order (*AR*) were applied, respectively. Coefficients *β* and α were estimated using the ordinary least squares technique, and the Durbin–Watson statistic was subsequently calculated. The sample period *t* comprised 55 monthly estimates, encompassing the forest fire season of August–December in 2009–2019.

The underlying assumption in Equation (1) was that the temporal impact of seasonal variations in meteorological conditions influences human activities. Specifically, the annual burning season usually takes place from August to December in the Legal Amazon [37]. This coincides with the dry and high-temperature periods when farmers commonly use fire as a means for land clearance, leading to an increased frequency of forest fires (Figure 2A) [37].

The foundational assumption in Equation (2) posited that variations in air quality have the potential to influence the distribution of allergens, consequently impacting the occurrence of respiratory diseases [6,38,39]. Substantial quantities of particulate matter, including PM_2.5_ and other pollutants, are emitted into the atmosphere during forest fires [14,17,25]. These particles can deeply infiltrate the lungs, resulting in respiratory issues and exacerbating pre-existing conditions like asthma, chronic obstructive pulmonary disease, or allergies (Figure 2B) [14].

All statistical analyses were conducted using R programming language version 4.3 (R Development Core Team, Vienna, Austria).

## 3. Results

Between 2009 and 2019, a staggering total of 1,438,322 wildfires were documented in the Legal Amazon region, with their occurrence estimated by the MODIS sensor aboard the AQUA_M-T satellite. The monthly distribution of forest fires exhibited a pronounced skew towards a concentrated period, primarily from August to December, encompassing 1,218,606 (85%) of all forest fires recorded during the August–December 2009–2019 period. Notably, the state of Pará experienced the highest incidence of wildfires during the forest fire season, comprising 30% (369,594) of the total series. Following closely were Mato Grosso with 18% (225,114), Maranhão with 18% (222,382), Tocantins with 10% (117,055), Amazonas with 8% (100,054), Rondônia with 8% (97,537), Acre with 5% (59,514), Amapá with 2% (20,451), and Roraima with 1% (6095).

From January 2009 to December 2019, the total number of hospital admissions for respiratory diseases among the selected age groups (0–14 years old and 60 years and above) in the Legal Amazon was 1,532,228. Two-thirds of these hospital admissions occurred during the months of the year where the climate is cooler (lower temperature) and wetter (more precipitation), often called the Amazonian rainy season. The remaining one-third (566,707) occurred in the forest fire season, from August to December 2009–2019. Pará emerged with the highest morbidity rate in the forest fire season, representing 33% (189,048) of the total respiratory-related hospital admissions, followed by Maranhão at 24% (138,538), Mato Grosso at 12% (67,604), Amazonas at 10% (55,391), Rondônia at 8% (43,898), Tocantins at 5% (30,259), Amapá at 3% (15,296), Acre at 3% (14,058), and Roraima at 2% (12,615).

In the forest fire season from August to December from 2009 to 2019, individuals between 1 and 4 years old constituted 34% (192,296) of respiratory-related hospital admissions, followed by those aged 60 and above at 29% (164,011) and infants under 1 year old at 18% (104,391). Admissions were relatively low in the age groups of 5 to 9 years (13%; 71,870) and 10 to 14 years (6%; 34,139). Monthly variations in hospital admissions for respiratory diseases showed a strong intercorrelation among age groups, indicating that an increase in cases among those 1 to 4 years old during a given month in the forest fire season is likely to coincide with an increase in cases among those aged 60 and above. The male vs. female ratio of the total number of admissions per age groups showed that males were generally most affected, primarily infants under 1 year old (ratio = 1.36; 60,223 vs. 44,168), from 1 to 4 years old (ratio = 1.21; 105,166 vs. 87,130), from 5 to 9 years (ratio = 1.15; 38,422 vs. 33,448), and those aged 60 and above (ratio = 1.10; 86,024 vs. 77,987).

Figure 3A presents the incidence of forest fires per 1000 km^2^, while Figure 3B illustrates the rate of hospital admissions for respiratory diseases among the selected age groups per 10,000 people in each state during the forest fire season.

Figure 3C depicts the utilization of Pearson correlation to examine the relationship between hospital admissions for respiratory diseases and forest fire occurrences in the Amazonian states under study. States experiencing a higher density of forest fire events also demonstrated an increased rate of hospital admissions due to respiratory diseases, and vice versa (Figure 3C). This observation suggests a positive and linear correlation of 37% (*r* = 0.3689, *df* = 7, *t* = 1.0503, *p* = 0.3285). Collectively, our interpretation from Figure 3 suggests a potential connection between fire events and hospital admissions, leading us to the subsequent step in the data modeling.

### Evaluating Hypotheses on the Associations between Forest Fires and Hospital Admissions

Hypothesis testing was conducted using the data encompassing the Legal Amazon region (Figure 4A). The principal component analysis generated three principal components from the original variables. Together, the first two components (PCA1 and PCA2) explained 97% of the variance in precipitation, temperature, and relative humidity (Table 1). Loadings for PCA1 showed that it represented a combination of low-precipitation, low-temperature, and low-humidity conditions (Figure 4B). For PCA2, precipitation had a negative loading, while temperature had a positive loading, indicating a contrast between precipitation and temperature, with higher precipitation and lower temperature on one end and vice versa (Figure 4C). By comparing the previous bar plots (Figure 4B,C) with Figure 4D, we can visually discern an increase in the number of forest fire events during drier months, peaking in September each year and declining towards the year’s end.

The first equation of the model, depicted in Figure 4E, illustrates the forest fire incidence (*I*) as a function of the meteorological conditions, represented by the principal component axes 1 and 2 (PCA1 and PCA2). The estimated equation is as follows: *I* = 12,737.97 + 6463.779 × PCA1 + 10,692.52 × PCA2. This model indicates that forest fire incidence is significantly influenced by meteorological conditions represented by PCA1 and PCA2, with no significant autocorrelation in the residuals. In other words, a one-unit increase in the eigenvalues of PCA1 and PCA2 corresponds to increases of up to 6463 and 10,652 forest fire events in a given month, respectively.

In the second equation of Figure 4E, approximately 66% of the variability in hospital admissions for respiratory diseases was explained by the model. This model estimates the coefficients of hospital admissions for respiratory diseases (*K*) as a function of forest fire events (*I*) per month. The estimated equation, *K* = 8934.4 + 0.02230239 × *I*, suggests that hospital admissions for respiratory diseases are influenced by the monthly number of forest fires. No significant autocorrelation in the residuals was observed in this model equation. Overall, the model indicates that an average of 22 hospital admissions for respiratory diseases can be expected per 1000 forest fires each month during the forest fire season in the Legal Amazon.

Figure 4F illustrates a monthly trend in hospital admissions for respiratory diseases, with peaks consistently occurring in August and September. This pattern notably corresponds with the heightened incidence of forest fires during these months. Subsequently, both hospital admissions and forest fire incidents gradually decline towards December, marking the conclusion of the forest fire season in the Legal Amazon region, 2009–2019.

## 4. Discussion

Upon analyzing the data, we observed a correlation between decreased precipitation and relative humidity, coupled with increased temperature levels, indicating a significant increase in forest fire occurrences. Additionally, this heightened frequency of forest fires coincided with an uptick in hospital admissions for respiratory diseases within the selected age groups (0–14 years old and 60 years and above) across the majority of the Legal Amazon region during the forest fire seasons of 2009–2019.

Several authors posit a reciprocal relationship between the occurrence of extreme climate events, such as El Niño, and the proliferation of fire hotspots [13,19,40]. Notably, the year 2010 stands out as the most extensive and severe due to the highest positive temperature anomalies recorded in the Pacific and Atlantic Oceans from 1903 to 2010 [40]. It coincided with the occurrence of the most intense El Niño in the last 70 years, coupled with unprecedented precipitation deficits across the entire Amazon basin, earning the designation of the century’s drought [40]. Similarly, the years 1997/98, 2005, and 2015/16 were identified as years of extreme drought in the Amazon linked to the El Niño phenomenon, characterized by significant forest fires [19]. From this standpoint, it can be inferred that, during El Niño years, the decrease in precipitation and rise in temperatures exert a substantial impact on fire dynamics, influencing factors such as their size, duration, expansion, and speed [41].

On the contrary, some authors contend that, even in years devoid of El Niño occurrences, the patterns of fire hotspots maintain proportionality [42]. Presuming that the La Niña phenomenon signifies a period of abundant rainfall, the prevalence of fire hotspots should substantially decrease. Throughout the research period (2001 to 2019), we observed a relatively consistent distribution between normal years and those marked by the presence of El Niño and La Niña events. This observation is closely tied to the prevalent practice of burning for land preparation, whether for cattle ranching or cultivation, in Brazil [19]. According to a report by the World Bank, China, India, and the United States lead as the primary countries employing fire for agricultural or pasture residue burning, with Brazil ranking fourth [19]. Additionally, Indonesia and the Russian Federation are also prominent users of fire for similar purposes [19].

The environmental policy in Brazil can be characterized as inconsistent, if not contradictory [6,7]. Despite the approval of a new Forest Code (Law 12651/2012) in 2012, the instances of wildfires persistently spread across a vast territory [43]. Despite real-time monitoring efforts in the Amazon, policies for control and prevention have remained insufficient [43]. At the beginning of Jair Bolsonaro’s presidency in 2019, there was a notable encouragement of violating environmental laws, coupled with a new anti-environmental discourse that favored large-scale deforestation and wildfires in Amazonian municipalities [6,7]. During this year, Brazil ignited an international diplomatic crisis centered around the country’s wildfires [19]. In August 2019, a Decree of Guarantee of Law and Environmental Order was enacted, authorizing the Armed Forces to conduct preventive and repressive actions against environmental offenses, specifically wildfires and forest fires, incurring a daily cost of approximately USD 300,000 [19].

The occurrence of these wildfires is linked to the regional human occupation pattern, driven by migratory surges for mining and/or opening agricultural frontiers [6,35]. Wildfires result in incomplete combustion, emitting carbon monoxide, particulate matter, and ash particles of various sizes [9]. Finer particles, prevalent in air polluted by wildfires, have a pronounced impact on the respiratory system [11]. Acute health effects are primarily localized to individuals near the burning area, ranging from intoxication to death by asphyxiation due to reduced oxygen levels [9]. Exposure to varying pollution levels can have other consequences, correlating with hospital admissions, medical consultations, school or work absenteeism, or mortality data [9,10,18,22,28,44].

The state of Amazonas, traditionally regarded as the most preserved within the Legal Amazon, has recently undergone substantial changes in land use due to intense human occupation, resulting in significant aerosol particle emissions, primarily attributed to biomass burning [45,46,47,48]. A study conducted over a decade ago (2002 to 2009) revealed relatively low concentrations of fine particulate matter [49]. In our more recent study covering the period from 2009 to 2019, we observed an association between hospital admissions and wildfire incidents. During the 2023 forest fire season, the capital of Amazonas, Manaus, found itself engulfed in dense gray smoke from numerous wildfires, fueled by activities such as road construction and illegal land clearing, resulting in severe air pollution (Figure 5A) [50]. Residents faced respiratory issues due to the deteriorating air quality, and even tropical fish were affected by the extreme drought and associated fires [51,52]. The smog, originating from both the city outskirts and intentional fires for land clearance, was exacerbated by the severe drought and high temperatures in the region (Figure 5A) [50]. We could additionally note that the Mamirauá Institute recorded water temperatures reaching 40 °C in Tefé—an Amazonas municipality situated 400 km away from Manaus city [53]. This increase in temperature contributed to the mortality of dolphins, fish, and alligators during the same period [53]. Finally, we display a typical perspective observed by a local resident during a period of forest fire in the Amazon, as illustrated in Figure 5B.

Socioeconomic disparities can significantly influence individuals’ ability to mitigate and cope with the health effects of wildfire exposure [54]. For instance, economic conditions may dictate housing quality, influencing the extent of indoor air quality protection during wildfires. Additionally, disparities in access to healthcare resources and information may contribute to variations in preventive measures and timely medical interventions.

The age groups of 1 to 4 years and 60 years and above exhibit heightened vulnerability to wildfire-related respiratory issues for various reasons [55,56]. Young children, with developing immune systems and smaller airway sizes, are more susceptible to respiratory irritants associated with wildfire smoke. Individuals aged 60 years and above often experience weakened immune systems, age-related decline in lung function, and pre-existing health conditions, rendering them more prone to severe health impacts when exposed to wildfire smoke. Additionally, both young children and elderly individuals may encounter challenges in evacuating or adopting protective measures during a wildfire event. Limited mobility and, in some cases, reduced cognitive abilities can consequently increase their vulnerability.

Not only are hospitalizations associated with wildfires, but mortality is also—this issue extends beyond Brazil, as the United States faces challenges with hospitalizations as well. In Brazil, respiratory deaths linked to daily exposure to fine particulate matter from wildfires totaled 31,287 from 2000 to 2016 [57]. Similarly, a notable surge in daily respiratory visits, especially among children aged from 0 to 5 years, was observed at Rady Children’s Hospital (RCH) in San Diego County during wildfires [58]. The impact of wildfire-specific fine particles on pediatric respiratory health led to a substantial 30.0% increase in visits in San Diego County from 2011 to 2017 [59]. Even relatively modest wildfires had significant health impacts, particularly affecting younger children [59].

The limitation of this study was the inability to assess the direct impact of smoke on hospital admissions at an individual level, as it operated as an analysis conducted at the community scale.

## 5. Conclusions

In this study, we discovered a correlation between climate factors and the incidence of wildfires. This, in turn, was found to be linked to hospital admissions for respiratory diseases in two age groups combined: children and young teenagers (0–14 years) and individuals aged 60 years or older.

The observed association rate was estimated at 22 hospital admissions for respiratory diseases in the selected age groups per 1000 wildfire incidents in the majority of the Legal Amazon states during the period from 2009 to 2019.

Considering these conclusions, the following recommendations are proposed:

(1) Promote and enforce sustainable land use practices to minimize the risk of wildfires, including the implementation of controlled burns, creation of defensible spaces, and establishment of firebreaks.

(2) Develop community-based health surveillance systems to monitor respiratory health indicators in the aftermath of wildfire events, enabling early intervention and support.

(3) Strengthen coordination among pertinent agencies, encompassing forestry, public health, and emergency services, to guarantee a unified and effective response to wildfires and associated health emergencies.

## Figures and Tables

**Figure 1 ijerph-21-00675-f001:**
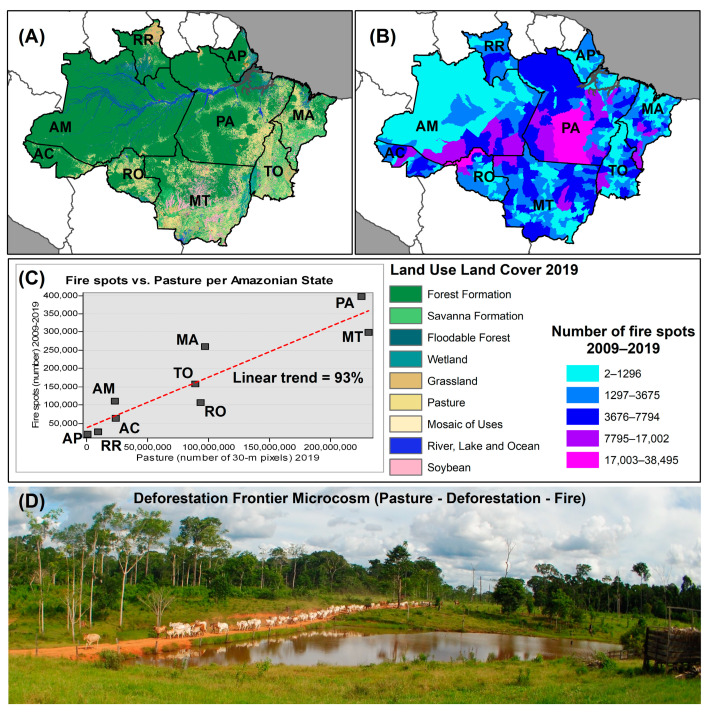
Study area and rational. (**A**) Land use land cover per state, Legal Amazon, 2019 [31]; (**B**) number of forest fire spots per municipality of states in the Legal Amazon, 2009–2019 [32]. Forest fire categories on the map were determined by natural breakpoints, where the difference in variation within categories is lower than the variation between categories; (**C**) scatter plot and linear trend of wildfire incidence case vs. pasture relative proportion per state (*r* = 0.9244, *df* = 7, *t* = 6.4147, *p* = 0.0004); and (**D**) microscale deforestation frontiers illustrate the correlation between deforestation fires and pasture lands. This image was taken by the senior author (G.Z.L.) using a Nikon D3000 camera (Nikon Corporation, Tokyo, Japan) in the rural settlement of Porto Dias, Acrelândia County, Acre state, 16 January 2015 (aperture and depth of field = f/10; shutter speed = 1/250 s; ISO speed = ISO-180).

**Figure 2 ijerph-21-00675-f002:**
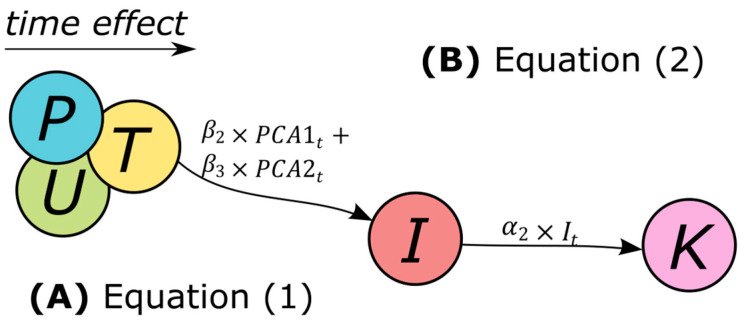
Temporal model of meteorological conditions, forest fires, and hospital admissions. (**A**) Equation 1 illustrates how fluctuations in precipitation, temperature, and relative humidity, as captured by the principal component analysis’ first axis (PCA1) and second axis (PCA2), influence the occurrence of forest fire events (*I*). (**B**) Equation 2 illustrates the effects of forest fires (*I*) on hospital admissions (*K*) in the forest fire season, August–December 2009–2019.

**Figure 3 ijerph-21-00675-f003:**
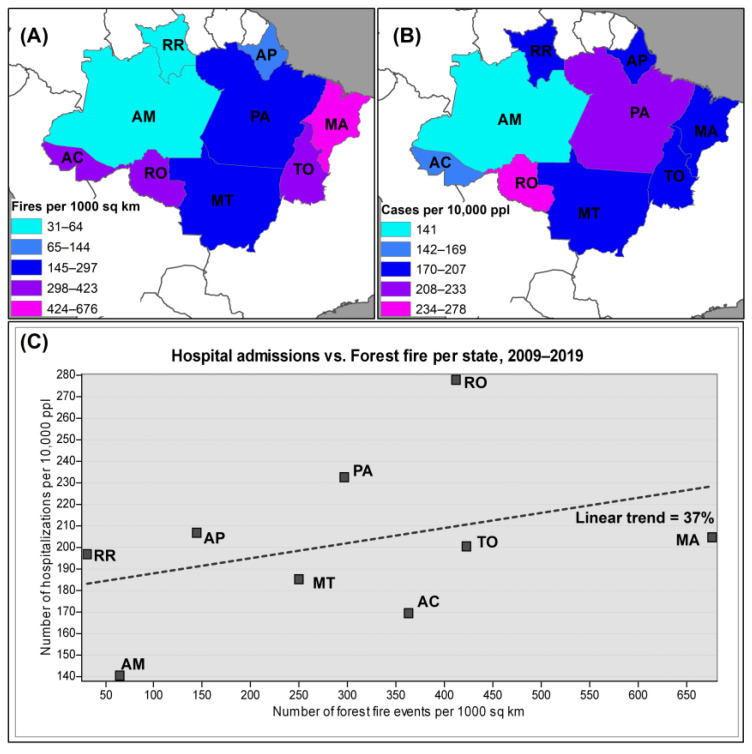
Initial data analysis in the forest fire season. (**A**) Forest fire events per 1000 square kilometers (sq km) of state area, Legal Amazon, August–December 2009–2019; (**B**) the rate of hospital admissions due to respiratory diseases among selected age groups per 10,000 people (ppl) in the state population of the Legal Amazon, August–December 2009–2019; and (**C**) a scatter plot and linear trend illustrating the incidence of wildfire area events against the rate of hospital admissions attributed to respiratory diseases in each state of the Legal Amazon region, August–December 2009–2019.

**Figure 4 ijerph-21-00675-f004:**
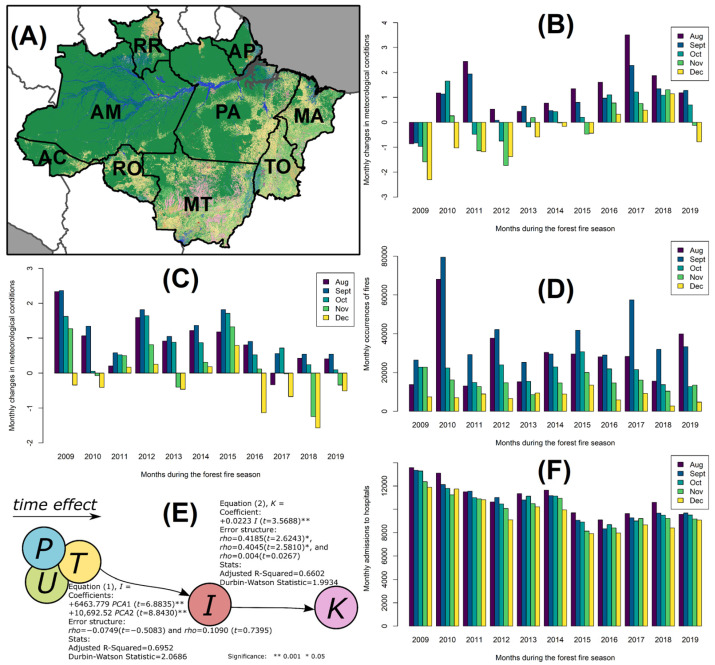
Temporal association of meteorological conditions, forest fire incidents, and hospital admissions for respiratory diseases in the selected age groups, Legal Amazon, August–December 2009–2019. (**A**) Nine states in the Legal Amazon; (**B**) PCA1 explains 67% of the variance in precipitation (loading = −0.5101), temperature (loading = −0.5171), and relative humidity (loading = −0.6872); (**C**) PCA2 explains 30% of the variance in precipitation (loading = −0.7131), temperature (loading = 0.7010), and relative humidity (loading = 0.0018); (**D**) monthly distribution of forest fire events in the forest fire season, 2009–2019; (**E**) results from the simultaneous equation models (Equations (1) and (2)) revealing coefficients which denote the strength and direction of the relationships between variables. The error structure accounts for adjustments made for temporal autocorrelation in the data. Key statistical metrics are provided for providing insights into the model’s performance. The Adjusted R-squared offers an evaluation of the model’s goodness of fit, with values ranging from poor (0) to excellent (1). The Durbin–Watson statistic aids in detecting the presence of autocorrelation, with values around 2.00 indicating no autocorrelation. Asterisks indicate statistical significance derived from hypothesis testing using *t*-tests; and (**F**) monthly distribution of hospital admissions for respiratory diseases in the selected age groups in the forest fire season, 2009–2019.

**Figure 5 ijerph-21-00675-f005:**
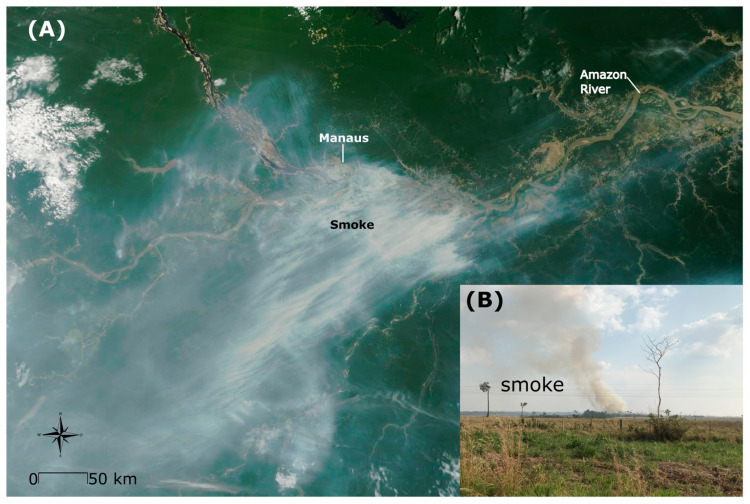
(**A**) An aerial image captured by the MODIS sensor on 11 October 2023, displays smoke streaming from fires burning near Manaus, state of Amazonas. Source: Publicly available image from NASA’s Earth Observatory [50]. (**B**) View of a forest fire. The first author (M.R.R.) captured this image with an iPhone 7 camera (Apple Inc, Cupertino, CA, USA) along the BR 364 Road near the city of Rio Branco in the Acre state on 28 August 2020. The camera settings for this shot were as follows: aperture and depth of field set at f/1.8, shutter speed at 1/4367 s, and ISO speed at ISO-20.

**Table 1 ijerph-21-00675-t001:** Numeric results pertaining to PCA analysis.

Rotation and Importance of Components	PCA1	PCA2
Precipitation (loading factor)	−0.5101571	−0.713132017
Temperature (loading factor)	−0.5171063	0.701027195
Relative humidity (loading factor)	−0.6872706	0.001896973
Explained variance proportion	0.6741	0.2976
Cumulative explained variance proportion	0.6741	0.9717

## Data Availability

Publicly available datasets were analyzed in this study. The original data presented in the study are openly available in [30,31,32,33,34].

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
