# Peer review of "Amazon Wildfires and Respiratory Health: Impacts during the Forest Fire Season from 2009 to 2019"

_ijerph, 2024, doi:10.3390/ijerph21060675_

Round 1
Reviewer 1 Report (Previous Reviewer 1)
Comments and Suggestions for Authors
The authors' response to the review comments overall insisted on their original approaches to data analysis and results presentation. Since the manuscript has not been revised significantly, I insist on my original decision to reject this manuscript.
Comments on the Quality of English LanguageThe authors rejected to revise their manuscript, and thus I reject to change my original decision of "reject."
Author Response
Reviewer #1
Dear Reviewer,
Thank you for evaluating our manuscript. We respectfully disagree with your assessment that we have adhered strictly to our original approaches to data analysis and presentation of results, without significant revisions to the manuscript.
We have conducted a thorough re-analysis of the data, focusing specifically on the forest fire season (August to December), during which the correlation between the number of forest fires and the incidence of respiratory diseases becomes readily apparent.
To streamline our approach, we simplified the model. Initially, we performed a principal component analysis (PCA) on the meteorological variables, resulting in two uncorrelated axes (PCA1 and PCA2). These axes were then utilized as independent variables in the first equation, where we examined their relationship with monthly forest fire counts. Subsequently, in the second equation, we directly correlated monthly forest fire counts with hospital admissions for respiratory diseases during the forest fire season.
We obtained monthly hospital morbidity data from the Department of Informatics of the Unified Health System, which is publicly accessible. Notably, we chose not to include a one-month latency in our models, resulting in their operation without any lag. Additionally, all references to lag have been eliminated from the previous version.
Furthermore, we have made corrections to Figure 3C, which now features a scatter plot and linear trend illustrating forest fire events per 1,000 square kilometers and the rate of hospital admissions due to respiratory diseases among selected age groups per 10,000 individuals.
Reviewer 2 Report (Previous Reviewer 2)
Comments and Suggestions for Authors
I thank the authors for their efforts in addressing my comments and suggestions.
Author Response
Reviewer #2
Dear Reviewer,
We appreciate your thorough review and value your feedback in enhancing the quality of our manuscript.
Reviewer 3 Report (New Reviewer)
Comments and Suggestions for Authors
Dear Editor Int. J. Environ. Res. Public Health
I have read the manuscript " Amazon Wildfires and Respiratory Health: Impacts During the Forest Fire Season (2009–2019)" and the following corrections are suggested:
-Remove the colon in the title
-Write the studied age group in the abstract
-provide a brief summary of the study method in abstract
-Superscript 2.5 in pm2.5
-Either list all abbreviations used or define all abbreviations once in the text. ppl.sq,...
- If there are studies that have determined the physical and chemical characteristics of particles and smoke resulting from fires, please include them in the introduction to better show the relationship between these pollutants and respiratory diseases.
-In the method of study, write the software used for statistical analysis
-Why are the numerical results related to PCA analysis not presented in a table?
-Since not all respiratory diseases are acute, why is there no lag time for fire and hospital admission?
Author Response
Reviewer #3
Dear Reviewer,
Thank you for your evaluation of our manuscript. We have reviewed the suggested corrections and responded as follows:
1) The title has been revised without the colon, now reading: “Amazon Wildfires and Respiratory Health: Impacts During the Forest Fire Season from 2009 to 2019”. (Lines 1-3)
2) The specified age groups were outlined in the abstract, as follows: “During the forest fire season, a substantial portion (566,707) of the total 1,532,228 hospital admissions for respiratory diseases in individuals aged 0–14 years and 60 years and above were recorded”. (Lines 23-25)
3) The methodology of the study has been incorporated into the abstract, outlined as follows: “A model consisting of two sets of simultaneous equations, with equation (1) illustrating seasonal fluctuations in meteorological conditions driving human activities associated with increased forest fires, and equation (2) representing how air quality variations impact the occurrence of respiratory diseases during forest fires.”. (Lines 25-29)
4) The abbreviation for fine particulate matter has been updated throughout the text, including within reference titles, and now appears as: PM2.5.
5) We have chosen to define all abbreviations once in the text and then use them freely thereafter. The previously undefined abbreviations in the legends of Figure 3 have been revised and now read as: “Initial data analysis in the forest fire season. (A) Forest fire events per 1,000 square kilometers (sq km) of state area, Legal Amazon, August–December 2009–2019; (B) The rate of hospital admissions due to respiratory diseases among selected age groups per 10,000 people (ppl) in the state population of the Legal Amazon, August–December 2009–2019”. (Lines 314-318)
6) To better illustrate the link between pollutants from fires and respiratory diseases, we incorporated findings from studies employing the Weather Research and Forecasting online-coupled Chemistry (WRF-Chem) model in the introduction, as follows: “Prior research has indicated that deforestation, precipitation, and temperature explained roughly 80% of the variability in forest fire seasons, highlighting a positive association be-tween fire count and deforestation [6,7]. The escalation of deforestation since 2012 resulted in a 39% increase in forest season fires in 2019, leading to an estimated 3,400 additional deaths due to heightened exposure to particulate air pollution [6,7].” (Lines 53-58).
7) The statistical analysis software was specified in the text as follows: “All statistical analyses were conducted using R version 4.3.”. (Line 275)
8) A table has been included to contain the numerical results pertaining to PCA analysis (Table 1) (Line 350)
Table 1. Numeric results pertaining to PCA analysis.
|
Rotation and importance of components |
PCA1 |
PCA2 |
|
Precipitation (loading factor) |
-0.5101571 |
-0.713132017 |
|
Temperature (loading factor) |
-0.5171063 |
0.701027195 |
|
Relative humidity (loading factor) |
-0.6872706 |
0.001896973 |
|
Explained variance proportion |
0.6741 |
0.2976 |
|
Cumulative explained variance proportion |
0.6741 |
0.9717 |
9) During the forest fire season, respiratory diseases tend to be acute. To capture this, we concentrated on this timeframe and utilized monthly hospital morbidity data in our models without any lag.
This manuscript is a resubmission of an earlier submission. The following is a list of the peer review reports and author responses from that submission.
Round 1
Reviewer 1 Report
Comments and Suggestions for Authors
This study has several methodological problems:
1) The lag time between a wildfire event and hospital admissions is typically within 5 days, according to previous studies. Thus, the lag time of 1 month in Equation (2) is not a reasonable assumption.
2) The models should use rates but not cases of hospital admissions, unless the populations were stable over the study period.
3) Why does Equation (2) include relative humidity, since it’s highly (reversely) related to temperature?
4) Figure 3C does not reveal the true relationship between hospital admissions and fire events. This linear relationship may be simply due to areas of the states, if their population densities and fire event densities are similar.
Author Response
Reviewer 1
This study has several methodological problems:
Response: We appreciate the chance to revise our manuscript.
1) The lag time between a wildfire event and hospital admissions is typically within 5 days, according to previous studies. Thus, the lag time of 1 month in Equation (2) is not a reasonable assumption.
Response: We acknowledge that the typical lag time between a wildfire event and hospital admissions is generally within 5 days. Nevertheless, there are reasons why a delay of about a month between a forest fire and an increase in respiratory hospitalizations may occur. This is substantiated by the understanding that the lag between a forest fire and a rise in respiratory hospitalizations can be influenced by various factors related to exposure, symptom onset, and healthcare-seeking behavior. Recognizing that the clarity of the explanation for this assumption was lacking in the previous version of the manuscript, we have introduced these additional paragraphs just before presenting the equations that illustrate the lagged terms:
A short-term association between a wildfire event and hospital admissions is anticipated; typically, the lag time is expected to be within 5-7 days, as indicated by previous studies conducted in Brazil and other locations [10,18,25]. Conversely, it is possible to discern underlying trends over prolonged durations. For example, an anticipated 30% in-crease in respiratory diseases hospitalizations annually is expected during drought years in Porto Velho, the third-largest city in the Amazon [17]. This standpoint reinforces the concept that the temporal gap between a forest fire and an uptick in respiratory hospitalizations can be influenced by various factors, as outlined below [9,11]:
Time for Exposure and Health Effects: Individuals may require time to be exposed to the smoke and pollutants released during a forest fire, with respiratory symptoms potentially taking days or weeks to manifest.
Cumulative Exposure: Prolonged or recurrent exposure to pollutants from a forest fire may contribute to the gradual development or exacerbation of respiratory conditions over time, resulting in a steady increase in hospitalizations rather than an immediate surge.
Latency in Health Effects: The development of specific respiratory diseases due to prolonged exposure to pollutants may not lead to hospitalization until the condition be-comes severe.
Healthcare-Seeking Behavior: Individuals experiencing respiratory symptoms may not immediately seek medical attention, influenced by underestimating symptom severity, lack of awareness, or delays in accessing healthcare services.
Considering the collective impact of these factors, we posit that there could be a delay of approximately one month between a forest fire and an increase in respiratory hospitalizations (Figure 2B).
2) The models should use rates but not cases of hospital admissions, unless the populations were stable over the study period.
Response: This approach has been extensively utilized and is supported by the stability of the population, with a key emphasis on the inclusion of specific error terms (temporal autocorrelation terms). These terms facilitate the adjustment of estimates to accommodate linear trends in cases linked to population growth. Nevertheless, we recognize that the initial version of the manuscript did not sufficiently clarify this rationale. As a result, we have introduced the following paragraph immediately following the presentation of the general equations (1) and (2), as follows:
The utilization of hospital admission cases, as opposed to rates (e.g., cases per 10,000 individuals), is justified by the stability of the population over the study period. The average population increase based on state data in the Legal Amazon region from the 2010 to the 2022 Census was approximately 10%. Additionally, the inclusion of the temporal autocorrelation error term in the models (equations 1 and 2) assists in correcting linear trends associated with population growth and refining the estimates.
3) Why does Equation (2) include relative humidity, since it’s highly (reversely) related to temperature?
Response: The incorporation of relative humidity in equation (2) is justified, given its substantial impact on respiratory health. While temperature directly influences the air’s water-holding capacity, changes in water vapor content can reciprocally affect relative humidity. Independent of temperature, multiple factors can alter water vapor content, including wind, proximity to water sources, vegetation and land use, and human activities. To provide clarity on the inclusion of relative humidity, a supplementary paragraph has been added after Figure 2B for the benefit of readers who may be inferring its significance.
The incorporation of relative humidity in equation (2) (Figure 2B) is justified due to its potential influence on respiratory health in multiple ways, including airway irritation, mucous membrane health, viral transmission, asthma symptoms, and inflammatory response [37]. While temperature directly affects the air’s ability to hold water vapor, changes in water vapor content can, in turn, impact relative humidity. Several factors, independent of temperature, can alter water vapor content, including wind, proximity to water sources, vegetation and land use, and human activities [38].
4) Figure 3C does not reveal the true relationship between hospital admissions and fire events. This linear relationship may be simply due to areas of the states, if their population densities and fire event densities are similar.
Response: The relationship depicted in Figure 3C may be subject to additional influencing factors, but the factors mentioned do not obscure the clear connection between fire events and hospitalizations. This is evident in the contrasting relationship between fire events and hospitalizations in states of comparable size and population. Conversely, states with substantial differences in size may exhibit similar levels of fire events and hospitalizations. To enhance clarity in interpreting Figure 3C, the following paragraph has been added just before the figure:
The depicted relationship in Figure 3C may be influenced by various factors, including the sizes of the states and their population, evident from the distances of each point in the graph relative to the trend line, indicating the average correlation. Nonetheless, it also illustrates the connection between fire events in Figure 3A and hospitalizations in Figure 3B. Larger states, such as AM and PA, exhibit varying numbers of hospitalizations, likely linked to the frequency of fire events. For example, despite PA and AM being nearly the same size, the former experiences the highest number of both fire events and hospitalizations, while the latter experiences the opposite. In contrast, AM and TO exhibit similar patterns of fire events and hospitalizations, despite notable differences in their territory sizes and populations.
Reviewer 2 Report
Comments and Suggestions for Authors
In reviewing the manuscript, I find your study both timely and significant. To enhance the clarity and depth of your manuscript, I recommend incorporating the following:
Please include a brief description of the hospital admissions data source. Details about data collection, scope, and any inherent biases would offer greater transparency and reliability to your research.
Please acknowledge any limitations in the hospital admission data, such as potential underreporting or regional data collection discrepancies.
Please reflect on the generalizability of your findings to other regions, considering Amazon's unique environmental and socio-economic contexts.
Also, please address limitations related to the temporal scope, including any influences of changing environmental policies or healthcare access over the study period.
Consider discussing how socioeconomic factors, including economic conditions and access to healthcare, may influence the vulnerability of populations to wildfire-related respiratory issues.
Please discuss in more depth why certain age groups (1 to 4 years and 60 years and above) are more impacted and how this compares with other age-related vulnerability studies in similar contexts.
Finally, an expanded conclusion on implications based on your findings would be highly valuable. This could include specific recommendations for forest management, public health strategies, and emergency preparedness.
Comments on the Quality of English Language
No Comments
Author Response
Reviewer 2
In reviewing the manuscript, I find your study both timely and significant. To enhance the clarity and depth of your manuscript, I recommend incorporating the following:
Response: Thank you for taking the time to review this manuscript. We appreciate your positive feedback. We carefully considered and incorporated the recommended improvements into our work.
Please include a brief description of the hospital admissions data source. Details about data collection, scope, and any inherent biases would offer greater transparency and reliability to your research.
Response: We value your suggestions. The revised version now incorporates this brief description immediately following the subsection on Hospital Admissions, as outlined below:
The DATASUS system aggregates data from a broad spectrum of healthcare institutions, including both public and private hospitals, ensuring comprehensive coverage of respiratory-related hospital admissions. The system captures essential information such as patient demographics, admission dates, diagnoses, and medical procedures. The dataset encompasses a diverse range of disorders such as asthma, chronic obstructive pulmonary disease, pneumonia, and other respiratory illnesses. With representation across various regions and states within Brazil, the dataset provides a geo-graphically inclusive perspective on respiratory health trends. It’s important to note that the data may be influenced by healthcare-seeking behavior, as individuals with varying severity of respiratory conditions may opt for different approaches in seeking medical attention.
Please acknowledge any limitations in the hospital admission data, such as potential underreporting or regional data collection discrepancies.
Response: Healthcare-seeking behavior invariably introduces bias in passive surveillance systems like DATASUS. This implies that the data may be influenced by individuals with respiratory conditions of varying severity opting for different patterns of seeking medical attention. This information has been incorporated into the brief description of this system:
It’s important to note that the data may be influenced by healthcare-seeking behavior, as individuals with varying severity of respiratory conditions may opt for different approaches in seeking medical attention.
Please reflect on the generalizability of your findings to other regions, considering Amazon's unique environmental and socio-economic contexts.
Response: In the initial manuscript, the following paragraph was written just before the conclusion of the Discussion Section:
Not only are hospitalizations, but mortality is also associated with wildfires—this issue extends beyond Brazil, as the United States faces challenges with hospitalizations as well. In Brazil, respiratory deaths linked to daily exposure to fine particulate matter from wildfires totaled 31,287 from 2000 to 2016 [54]. Similarly, a notable surge in daily respiratory visits, especially among children aged 0 to 5 years, was observed at Rady Children’s Hospital (RCH) in San Diego County during wildfires [55]. The impact of wildfire-specific fine particles on pediatric respiratory health led to a substantial 30.0% increase in visits in San Diego County from 2011 to 2017 [56]. Even relatively modest wildfires had significant health impacts, particularly affecting younger children [56].
This paragraph addresses the generalizability of our findings to other regions, referencing a study conducted in San Diego, California, US, which exhibited similar trends to ours.
Also, please address limitations related to the temporal scope, including any influences of changing environmental policies or healthcare access over the study period.
Response: In the original manuscript, we mentioned that there was a shift in environmental policies in Brazil during the last year of our time-series data. While this shift did not impact our analysis, we addressed it in the fourth paragraph of the Discussion Section:
At the beginning of Jair Bolsonaro’s presidency in 2019, there was a notable encourage-ment of violating environmental laws, coupled with a new anti-environmental discourse that favored large-scale deforestation and wildfires in Amazonian municipalities [6,7]. During this year, Brazil ignited an international diplomatic crisis centered around the country’s wildfires [19]. In August 2019, a Decree of Guarantee of Law and Environmental Order was enacted, authorizing the Armed Forces to conduct preventive and repressive actions against environmental offenses, specifically wildfires and forest fires, incurring a daily cost of approximately US$300,000 [19].
Consider discussing how socioeconomic factors, including economic conditions and access to healthcare, may influence the vulnerability of populations to wildfire-related respiratory issues.
Response: We have incorporated a new paragraph into the Discussion Section immediately following Figure 5 to provide a more comprehensive discussion on the connection between socioeconomic disparities and respiratory issues arising from wildfires:
Socioeconomic disparities can significantly influence individuals’ ability to mitigate and cope with the health effects of wildfire exposure [54]. For instance, economic conditions may dictate housing quality, influencing the extent of indoor air quality protection during wildfires. Additionally, disparities in access to healthcare resources and information may contribute to variations in preventive measures and timely medical interventions.
Please discuss in more depth why certain age groups (1 to 4 years and 60 years and above) are more impacted and how this compares with other age-related vulnerability studies in similar contexts.
Response: This recommendation holds significant importance. To address this, we have introduced a new paragraph into the Discussion Section, outlined as follows:
Age groups 1 to 4 years and 60 years and above exhibit heightened vulnerability to wildfire-related respiratory issues for various reasons [55,56]. Young children, with developing immune systems and smaller airway sizes, are more susceptible to respiratory irritants associated with wildfire smoke. Individuals aged 60 years and above often experience weakened immune systems, age-related decline in lung function, and pre-existing health conditions, rendering them more prone to severe health impacts when exposed to wildfire smoke. Additionally, both young children and elderly individuals may encounter challenges in evacuating or adopting protective measures during a wildfire event. Limited mobility and, in some cases, reduced cognitive abilities can consequently increase their vulnerability.
Finally, an expanded conclusion on implications based on your findings would be highly valuable. This could include specific recommendations for forest management, public health strategies, and emergency preparedness.
Response: Certainly. We have incorporated the following into the Conclusions section:
Considering these conclusions, the following recommendations are proposed:
(1) Promote and enforce sustainable land use practices to minimize the risk of wild-fires, including the implementation of controlled burns, creation of defensible spaces, and establishment of firebreaks.
(2) Develop community-based health surveillance systems to monitor respiratory health indicators in the aftermath of wildfire events, enabling early intervention and sup-port.
(3) Strengthen coordination among pertinent agencies, encompassing forestry, public health, and emergency services, to guarantee a unified and effective response to wildfires and associated health emergencies.